# Changes in Healthy Behaviors among Arab Israeli Children Diagnosed with ASD amid the Coronavirus Outbreak: Mothers' Perceptions

Rafat Ghanamah 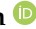

Special Education Department, Sakhnin College for Teacher Education, Sakhnin 3081000, Israel; rafatgan1980@sakhnin.ac.il

**Abstract:** The constraints imposed by the COVID-19 pandemic may have adverse effects on the health behaviors of children and adolescents, particularly those with neurodevelopmental conditions such as autism spectrum disorder (ASD). This study aimed to examine changes in health-related behaviors, including physical activity, screen time, and sleep duration, among children with ASD and their adherence to the 24-h movement guidelines during the pandemic. An online cross-sectional survey was conducted with 46 Arab Israeli mothers of children diagnosed with ASD. According to the responses provided by the mothers, the findings indicate a noteworthy reduction in physical activity, a marked rise in screen time, and a considerable increase in sleep duration amid the COVID-19 pandemic. Additionally, there was a decline in the portion of the sample complying with physical activity and screen time recommendations, coupled with an increase in the percentage of children meeting sleep duration guidelines. The prevalence of ASD children adhering to the overall 24-h movement guidelines was notably low during the COVID-19 outbreak. These findings contribute to the growing body of literature indicating adverse effects of the coronavirus pandemic on individuals with neurodevelopmental disorders, emphasizing the urgent necessity for healthcare, interventions, and programs tailored to ASD children.

**Keywords:** Arab Israeli children; ASD; coronavirus; physical activity; screen time; sleep duration; education

## 1. Introduction

The new coronavirus, officially designated as COVID-19, initially emerged in China in December 2019 and rapidly spread worldwide. Recognizing its widespread impact, the World Health Organization (WHO) officially declared it a global pandemic on 11 March 2020 [1]. Individuals afflicted by the coronavirus may subsequently manifest challenging respiratory symptoms, with transmission occurring through both direct contact and respiratory droplets. As of 12 October 2023, the cumulative toll has resulted in approximately seven million fatalities and 771.2 million documented cases of COVID-19 on a global scale [2].

On 27 February 2020, Israel recorded its first case of the coronavirus. Since then, the ensuing pandemic has resulted in approximately 4.9 million confirmed positive cases and 12,691 fatalities in Israel as of the latest update on 16 October 2023 [2]. In response to the crisis, Israeli authorities implemented three comprehensive lockdowns, spanning from March 2020 to the beginning of 2021. These measures were instituted to mitigate the spread of the pandemic, enforce social isolation, and restrict movement. Consequently, educational institutions, including schools and academic facilities, were temporarily closed. Traditional classrooms were replaced with small-group learning arrangements, and the predominant mode of instruction transitioned to online and at-home distance learning [3].

Related to the current research, the Arab population in Israel, numbering nearly 2.04 million individuals and constituting 21.1% of the total Israeli citizenry, represents a distinctive national and linguistic minority group within the country [4]. Approximately 85% of the Arab population in Israel identifies as Muslim Arabs, while 7.4% are Druze, and

7.2% are Christian Arabs, according to the Central Bureau of Statistics (CBS) 2019. Among the Muslim Arab population, 16% are Bedouins, comprising about 3.5% of the total Israeli population. Furthermore, nearly 90% of the Arab population resides in predominantly Arab towns, primarily located in Northern Israel. The Negev region in the south of Israel is home to two-thirds of the Bedouin population, with some residing in unrecognized settlements lacking basic infrastructure, such as electricity, sewage systems, and telephone lines (as reported by CBS 2019). In the past, the Arab population in Israel shared various demographic factors, such as lower economic means, larger family sizes, densely populated living conditions, cultural distinctions, peripheral geographical location, and an increased propensity for comorbidities [5–8]. A study conducted by [5] revealed significant influence of the patriarchal worldview on various aspects of Arab society, including social, cultural, economic, and political practices. The research highlighted the prevailing dominance of extended family structures over liberal Western values, which typically emphasize objectivity and individualism. It is noteworthy that, within Arab society, where the extended family holds considerable importance rooted in patriarchal values, women are largely underrepresented in political, cultural, and economic spheres [5].

Nevertheless, in recent decades, the Arab Israeli community has experienced significant economic, political, and social transformations, resulting in shifts in traditional family dynamics and a leaning towards more Western, individualistic lifestyles. Presently, many Israeli Arab nuclear families exhibit a blend of modern and traditional elements, with the latter typically holding greater sway [9] (Blit-Cohen and Jammal-Abboud, 2017). These changes are not common and depend on the region and religion (northern areas and the Christian population are more open societies and individualistic).

The COVID-19 outbreak has markedly affected children with special needs, as the policy of physical distancing has resulted in a substantial deprivation of opportunities for these individuals to enhance their well-being. The extensive disruption of medical care and rehabilitation services during the lockdown, coupled with the challenges of physical communication crucial for their mental and physical development, has posed significant obstacles [10]. The consequences of the lockdown have been particularly pronounced, negatively impacting various facets of approximately 50% of children with special needs. These include sensory–motor development, cognitive functioning, sleep patterns, behavior, morale, and social relationships. Moreover, there exists convincing evidence highlighting the adverse effects of both COVID-19 and the preventive measures implemented to curb its spread on families with children possessing disabilities [10,11].

Children with disabilities and special needs may also be more susceptible to getting the virus and developing problems due to specific primary medical disorders. These children, especially those with sensory processing and integration issues like tactile, vestibular, proprioceptive, and difficulties with hearing, vision, and cognitive performance, have difficulty taking the necessary precautions during epidemic times, like donning masks, maintaining physical distance, etc. [12]. Autism is a spectrum disorder (ASD) with a wide variety of clinical symptoms. Autism itself places a significant burden on families of children with autism, with severe economic and psychological burdens. This impact has been multiplied many times over by the coronavirus outbreak [13–15].

Children and adolescents with autism often suffer from co-occurring psychiatric conditions. Autism is characterized by social and communication difficulties, as well as restricted and repetitive behaviors, interests, and activities [16]. Many children and adolescents with autism rely on carefully constructed support networks and daily routines that were abruptly interrupted by the onset of the COVID-19 pandemic. The first coronavirus lockdown in Israel began in March 2020. School buildings were closed, classes for most students quickly moved to online learning, and all but essential services were forced to close. This major change in daily life can have an impact on the mental health of young people with autism and their families [17].

The majority of children with ASD stopped obtaining crucial education and clinical treatments during the closure. Additionally, children with ASD often resist deviations

from their routines. Therefore, most of them were forced to endure confinement when kindergartens, schools, and other health services that they usually attended daily were closed. Simultaneously, the family shows alterations in its structure, with parents often absent from home, spending more time with siblings, or separated from their usually present grandparents [10]. Additionally, children with ASD or intellectual disabilities are more likely to be neglected by others throughout the pandemic, whilst basic caring public supports are no longer applicable [18]. Moreover, prolonged stay at home increases inattentive and hyperactive behaviors, screen and game addiction, and sleep disturbances, leading to accompanying mental health disorders and reduced healthy behaviors in autistic children. Depriving autistic children of therapeutic intervention will cause an environmental deprivation of the specific tools, devices, and sensory inputs that accelerate changes in development and the rate of progress. This change is necessary because online education and home-based training alone cannot overcome clinical symptoms in autistic children [19].

For children and adolescents with ASD, growing research shows that increased participation in physical activity (PA), reduced screen time (ST), and appropriate sleep duration can help improve symptoms associated with ASD and have the potential to alleviate the stress and anxiety associated with the coronavirus pandemic [20–22]. Unfortunately, there is a potential that the COVID-19 outbreak will negatively affect people's health-related activities and the behavioral aspects of children and adolescents with ASD, potentially creating barriers to PA engagement, while also increasing rates of sedentary behavior and screen time due to being at home. As recent studies continue to demonstrate the advantages of participating in PA, including reduced anxiety and improved social skills and communication problems, children and adolescents with ASD must continue to participate in PA, while limiting ST. Additionally, reduced PA levels, along with increased ST levels, may adversely impact sleep patterns, which can be especially challenging for people with ASD, who are already prone to sleep difficulties [23–25].

It is crucial to investigate whether the COVID-19 pandemic may have an impact on health-related behaviors, given the magnitude of health-related behaviors, e.g., physical activity and screen time, in children and youth with ASD. Furthermore, children from marginalized populations, including ethnic minorities, and those with pre-existing mental challenges, are particularly susceptible to experiencing heightened vulnerabilities due to imposed restrictions [7,8]. With this in mind, the present study seeks to explore the impact of the COVID-19 pandemic on the health behaviors of Arab Israeli children with autism spectrum disorder (ASD), encompassing physical activity (PA), screen time (ST), and sleep duration. The investigation also aims to discern potential variations across age groups, specifically focusing on preschool- and primary-school children.

An additional objective of this study is to illuminate the impacts of the lockdown and "stay home" directives on children with ASD, contributing valuable insights to the existing knowledge base regarding the effects of the COVID-19 outbreak on individuals with neurodevelopmental disorders, such as children with ASD. Notably, as of 2021, nearly 37 thousand children and adolescents (up to 18 years old) in Israel had received an ASD diagnosis, with approximately 10% of them belonging to the Arab population.

To act as a comparative control in our study, a recent investigation conducted by [7] investigated changes in physical activity, length of sleep, and screen time amongst 490 Arab Israeli children with typical development, based on their parents' perceptions, during the COVID-19 restrictions. The findings revealed noteworthy changes during the pandemic, including a decrease in time spent engaging in physical activity and an increase in both screen time and sleep duration, coupled with a decline in the percentage of children adhering to the suggested recommendations for physical activity and screen time. Moreover, the study highlighted that only a small proportion of participants achieved the overall 24-h movement guidelines during the pandemic. Significantly, schoolchildren demonstrated superior compliance with the recommended guidelines for PA and sleep length in contrast to preschoolers. These insights from the control group contribute valuable context for

understanding the potential impact of COVID-19 restrictions on the health behaviors of children with ASD in our specific study.

In the present study, we assessed healthy behaviors based on the 24-h movement guidelines, a framework originally developed in Canada. These guidelines represent a fundamental shift in the conceptualization of movement behaviors, moving away from a singular focus on specific types of activity, such as moderate-to-vigorous physical activity. Instead, they present a holistic view of combined movement behaviors. The guidelines offer recommendations for a complete 24-h period, taking into account behaviors linked to health outcomes. This includes not only the promotion of moderate-to-vigorous physical activity (at least 60 min per day) but also considerations for leisure screen time (limited to no more than 2 h per day) and the attainment of sufficient sleep duration (ranging from 9 to 11 h per night for children aged 5–13 years and 10–13 h for 3–5-year-olds).

Building upon the recent literature, our hypothesis posits that children with ASD experienced a more pronounced reduction in physical activity (PA), an increase in screen time (ST), and shorter sleep duration as a consequence of the COVID-19 pandemic and associated lockdowns. Additionally, we anticipate that children aged 6–10 years are more adversely affected than their younger counterparts (preschool children) concerning sleep duration, screen time, and physical activity during the coronavirus pandemic; younger children may be more protected by their parents than older children, and parents may provide nurturing care for younger children. This hypothesis is grounded in the understanding that the pandemic and associated restrictions may have distinct impacts on different age groups within the pediatric ASD population, with potential variations in how they engage in and are influenced by health-related behaviors.

By investigating whether such alterations occur, professionals and researchers can acquire valuable insights that may inform the development of interventions aimed at promoting healthy behaviors over the long term. This is particularly crucial during periods when children find themselves outside of structured environments and constant daily routines, such as the ongoing COVID-19 pandemic or times of conflict, like war. Understanding the impact of disruptions on health-related behaviors in these contexts allows for the identification of strategies and interventions to support and enhance the well-being of children facing such challenges.

## 2. Materials and Methods

### 2.1. Participants and Setting

The present research involved mothers of Arab Israeli children diagnosed with ASD, utilizing a digital survey instrument administered between 2 February and 28 February 2021. This timeframe followed the conclusion of three general lockdowns in Israel and extended periods of home quarantine. The collection of primary data from mothers employed non-probability sampling techniques. To recruit participants, mothers residing in Israel with a single child diagnosed with ASD, aged between three and ten years (preschool through fourth grade), were targeted. The corresponding author reviewed the diagnosis that was conducted by certified developmental psychologists. Inclusion criteria stipulated that the children did not exhibit severe physical, visual, or hearing difficulties that would impede their participation in physical activity or the use of screens. All participants were native speakers of Arabic and spoke the same local vernacular. The participants were recruited from the same geographical region (north Israel), with low-to-middle socioeconomic backgrounds. The socioeconomic status of the geographical region where the children of the current study come from is the same as that of most Arab towns and villages in Israel (The Israel Democracy Institute, West Jerusalem, Israel). Mothers meeting these criteria were identified through professional networks, teacher groups, and parent communities and were provided with a link to a Google form specifically designed for the survey.

The link was distributed across various social media platforms, including Facebook, Telegram, WhatsApp, and Instagram. Alongside the link, there was an audio clip and

written content explaining the study and its objectives. We utilized diverse recruitment channels, such as the Israeli Association of Autistic Subjects' Parents and institutions catering to children with ASD, to enrich our participant pool. Ultimately, 51 mothers submitted the forms to the authors. Following the exclusion of incomplete forms and inaccurate responses, a total of 46 participants were included in the final analysis.

The determination of the sample size was conducted before participant recruitment, employing a sensitivity power analysis through G*Power 3.1 [26]. The a priori analysis considered a significance level ($\alpha$) of 0.05 and a power of 0.90, with an effect size of 0.50. The results indicated that a sample size exceeding 15 participants per group was necessary for the assessment of interactions in the overall analysis. This encompassed both age categories (three-to-five years and six-to-ten years) across two distinct time points: pre-COVID-19 outbreak and during the COVID-19 outbreak.

Participants did not receive any financial incentives, and the confidentiality of their information was assured. Informed consent was sought from all participants, and they were explicitly informed that they could cease their contribution to the study at any point. The online study adhered to the guidelines outlined in the Declaration of Helsinki, and the local Institution's human research procedures of the (12/2020-103) were strictly followed.

### 2.2. Data Collection and Analyses

To adhere to COVID-19 pandemic restrictions, data collection was carried out using an electronic survey, and in-person interviews were avoided. This approach ensured the safety and well-being of both participants and researchers during the ongoing public health crisis.

#### 2.2.1. Sociodemographic Characters of the Sample

The survey incorporated sociodemographic questions pertaining to the mothers, including details such as educational level, gender, and age. The demographic information concerning the children encompassed age, gender, and school type. The classification of school type was delineated by age and grades, distinguishing between preschool (ages 3–5) and elementary school (grades 1 to 4/6–10 years). Furthermore, school type was categorized based on specific educational conditions, such as enrollment in a regular class, a special class within a regular school, or attendance at a special school. This comprehensive set of sociodemographic variables aimed to capture a nuanced understanding of the study participants and their educational contexts. Table 1 provides an overview of the sociodemographic characteristics of both mothers and children.

#### 2.2.2. Health Behaviors Measures

Mothers participated in a survey that included items aimed at assessing the health behaviors of their children both prior to and amid the coronavirus pandemic:

Physical activity (PA) was assessed using a single question: "Per week, how many days does your child participate in a minimum of 60 min of physical activity?" This measure has been demonstrated to possess satisfactory reliability and validity [27]. Respondents could choose from a range of options, spanning from zero to seven days weekly with increments of 1 day. Meeting the physical activity guidelines entails engaging in moderate-to-vigorous physical activity (MVPA) for a minimum of sixty minutes each day.

Screen time (ST) was evaluated by asking, "How many hours of screen time (tablet, iPhone, TV, computer, etc.) does your child spend on a standard weekday?" This question was also posed for a typical weekend. Notably, participants were explicitly instructed not to include time spent on electronic devices for school, including tablets and computers. Screen time was categorized based on guidelines: for children and adolescents, meeting ST guidelines meant spending less than 2 h per day, while for preschoolers, meeting the guidelines involved less than 1 h per day.

Sleep duration (SD) was assessed by individually querying mothers about their children's typical weekday and weekend bedtimes, as well as wake-up times. These inquiries were also posed about the coronavirus closure era. To calculate the mean of each day sleep

length for each child, the method proposed by Lopez-Gil et al. (2021) [28] was employed: [(average nighttime sleep length on weekdays × 5) + (average nighttime sleep length on weekends × 2)]/7. Aligned with global guidelines for early childhood development from the World Health Organization, responses falling within the ranges of 10–13 h for 3–5-years old and 9–11 h for 5–13 years old were categorized as "meeting sleep guidelines". Participants who did not select one of these options were regarded as "not meeting sleep guidelines" [29].

**Table 1.** Sociodemographic characteristics of mothers and children.

| | Total (N = 46) |
|---|---|
| **Mothers** | |
| Age {M (SD)} | 37.65 (6.22) |
| Age group {N(%)} | |
| 25 years old and below | 3 (6.5) |
| 26–35 years old | 7 (15.2) |
| 36–45 years old | 32 (69.6) |
| 46–55 years old | 3 (6.5) |
| Over 55 years old | 1 (2.2) |
| Education {N(%)} | |
| Elementary | 6 (13) |
| Secondary | 13 (28.3) |
| BA degree | 17 (37) |
| ≥MA degree | 10 (21.7) |
| Working (yes) {N(%)} | 31 (67.4) |
| Was one of your family members diagnosed with COVID-19? (yes) {N(%)} | 22 (47.8) |
| Was one of your family members in isolation? (yes) {N(%)} | 36 (78.3) |
| **Children** | |
| Female {N(%)} | 13 (28.3) |
| Age {M (SD)} | 6.37 (2.03) |
| Age level {N(%)} | |
| 3–5 years old | 23 (50) |
| 6–10 years old | 23 (50) |
| Type of school {N(%)} | |
| Regular kindergarten | 11 (23.9) |
| Kindergarten for children with special needs | 12 (26.1) |
| Regular class in regular school | 5 (10.9) |
| Special class in a regular class | 9 (19.6) |
| Special school | 9 (19.6) |
| Diagnosed with COVID-19 | 12 (26.1) |
| Asked to be in isolation | 25 (54.3) |

Descriptive statistics were utilized to characterize the demographics of both mothers and children. The proportion of children meeting the 24-h guidelines was calculated. A repeated measures analysis of variance (rm-ANOVA) was employed to examine differences between the pre-pandemic and during the pandemic periods with respect to PA, ST, SD, and compliance with the three 24-h movement guidelines. A similar procedure was undertaken to compare two age groups (3–5 years old and 6–10 years old) concerning physical activity, screen time, sleep duration, and meeting the three 24-h movement guidelines, both before and throughout the coronavirus pandemic. In instances where rm-ANOVA indicated a

significant interaction, the basis of the interaction was determined using independent sample *t*-test analysis, with Cohen's d effect size computed for the *t*-test results.

All analyses were conducted using IBM SPSS Statistics 28, with a significance level set at $p < 0.05$.

## 3. Results

### 3.1. Sociodemographic Descriptions

In terms of the mothers, the respondents' ages ranged from 21 to 58, with the majority falling within the 36–45 age group (32 mothers, 69.6%). Additionally, 67.4% of the mothers were employed during the lockdown, and 37% held bachelor's degrees. (Refer to Table 1 for comprehensive details.)

Concerning the children, the demographic breakdown revealed that 71.7% were males, 50% fell within the age range of 3–5 years, 26.1% had tested positive for COVID-19 infection, and 54.3% were required to undergo isolation due to exposure to a person diagnosed with COVID-19 infection. (Refer to Table 1 for detailed information.)

### 3.2. Health-Related Behaviors Prior to and throughout the COVID-19 Outbreak

Table 2 presents the physical activity, screen time, and sleep duration of children prior to and through the coronavirus pandemic, categorized by the overall sample and age group.

**Table 2.** Means and standard deviations of children's physical activity, screen time, and sleep duration before and during the coronavirus outbreak (overall sample and by age group).

| Measure | Total (*n* = 46) | | 3–5 Years Old (*n* = 23) | | 6–10 Years Old (*n* = 23) | |
|---|---|---|---|---|---|---|
| | Pre M (SD) | During M (SD) | Pre M (SD) | During M (SD) | Pre M (SD) | During M (SD) |
| Physical activity (days a week) | 5.96 (0.76) | 4.41 (0.86) | 5.91 (0.79) | 4.48 (0.89) | 6 (0.74) | 3.34 (0.83) |
| Screen time (hours a day) | 1.61 (0.48) | 3.01 (0.65) | 1.58 (0.47) | 2.93 (0.71) | 1.65 (0.49) | 3.09 (0.59) |
| Sleep duration (hours a day) | 8.54 (0.55) | 9.35 (0.68) | 8.48 (0.60) | 9.15 (68) | 8.61 (0.50) | 9.55 (0.66) |

#### 3.2.1. Physical Activity

The analysis indicates a decline in physical activity (PA), with a significant main effect of time-point ($F_{(1, 44)}$ = 210.84, $p < 0.001$, η2 = 0.83). However, there was a non-significant interaction between physical activity and age group ($F_{(1, 44)}$ = 1.05, $p = 0.312$, η2 = 0.02), and there was no significant main effect of age group ($F_{(1, 44)}$ = 0.01, $p = 0.962$, η2 = 0). The two age groups did not exhibit significant differences before or during the COVID-19 outbreak (t (44) = −0.39, $p = 0.702$, d = 0.11, and t (44) = 0.46, $p = 0.649$, d = 0.14, respectively). These findings are noteworthy, indicating that, during the coronavirus pandemic, both age groups experienced a significant decrease in PA ($F_{(1, 22)}$ = 66.55, $p < 0.001$, η2 = 0.75, and $F_{(1, 22)}$ = 191.37, $p < 0.001$, η2 = 0.90; for children aged 3–5 and 6–10, respectively).

#### 3.2.2. Screen Time

During the COVID-19 pandemic, there was a noteworthy increase in screen time (ST) ($F_{(1, 44)}$ = 273.74, $p < 0.001$, η2 = 0.86). However, neither the main effect of age group ($F_{(1, 44)}$ = 0.62, $p = 0.437$, η2 = 0.01) nor the interaction between ST and age group ($F_{(1, 44)}$ = 0.19, $p = 0.664$, η2 = 0.00) reached statistical significance. The two age groups did not exhibit significant differences both before and during the COVID-19 outbreak (t (44) = −0.79, $p = 0.436$, d = 0.23; t (44) = −0.55, $p = 0.582$, d = 0.16). During the coronavirus outbreak, screen time levels significantly increased for both age groups ($F_{(1, 22)}$ = 137.91, $p < 0.001$, η2 = 0.86 and $F_{(1, 22)}$ = 136.13, $p < 0.001$, η2 = 0.86; for 3–5 and 6–10-year-olds, respectively).

### 3.2.3. Sleep Duration

Throughout the COVID-19 outbreak, the sleep duration significantly increased ($F_{(1, 44)} = 273.74$, $p < 0.001$, $\eta2 = 0.86$). However, neither age group nor sleep duration showed significant interactions ($F_{(1, 44)} = 2.23$, $p = 0.143$, $\eta2 = 0.05$) or main effects ($F_{(1, 44)} = 1.52$, $p = 0.225$, $\eta2 = 0.03$). The two age groups did not differ significantly before the COVID-19 outbreak ($t_{(44)} = -0.24$, $p = 0.824$, $d = 0.07$), and they also did not exhibit significant differences during the coronavirus pandemic ($t_{(44)} = 2.63$, $p = 0.019$, $d = 0.51$). Notably, between the ages of three and ten, both age groups slept more throughout the coronavirus pandemic than their usual patterns ($F_{(1, 22)} = 11.02$, $p = 0.003$, $\eta2 = 0.33$ and $F_{(1, 22)} = 37.95$, $p < 0.001$, $\eta2 = 0.63$, for the respective age groups).

### 3.3. Prevalence Rates of Meeting 24-Hour Guidelines

Table 3 presents the percentage of children meeting the 24-h movement guidelines.

**Table 3.** Prevalence rates of children meeting the 24-h guidelines.

|  | Total % (*n* = 46) | 3–5-Year-Olds % (*n* = 23) | 6–10-Year-Olds % (*n* = 23) | *t* (*p*, *d*) |
|---|---|---|---|---|
| Before COVID-19 outbreak |  |  |  |  |
| PA | 26.1 | 26.1 | 26.1 | 0 (1, 0) |
| ST | 67.4 | 34.8 | 100 | **−6.42 (<0.001, 1.89)** |
| SD | 52.2 | 4.3 | 100 | **−22 (<0.001, 6.49)** |
| 24 h combined | 15.2 | 4.3 | 26.1 | **−2.11 (0.041, 0.62)** |
| During COVID-19 outbreak |  |  |  |  |
| PA | 6.5 ↓ | 8.7 ↓ | 4.4 ↓ | −0.59 (0.561, 0.17) |
| ST | 6.5 ↓ | 0 ↓ | 13.4 ↓ | **−1.82 (0.038, 0.54)** |
| SD | 58.7 ↑ | 17.4 ↑ | 100 | **−10.22 (<0.001, 3.01)** |
| 24 h combined | 2.2 ↓ | 0 | 4.4 ↓ | −1 (0.323, 0.29) |

PA = physical activity, ST = screen time, SD = sleep duration, and 24 h combined = meeting the three 24-h movement guidelines. Values marked in bold indicate significance; ↑ = significant upsurge ($p < 0.001$), and ↓ = significant decline ($p < 0.001$).

The prevalence rates of children meeting the physical activity (PA) and screen time (ST) guidelines decreased during the COVID-19 outbreak ($F_{(1, 44)} = 10.73$, $p = 0.002$, $\eta2 = 0.20$; and $F_{(1, 44)} = 95.82$, $p < 0.001$, $\eta2 = 0.69$). The children exhibited a significant decline in PA ($p = 0.043$ and $p = 0.022$; 3–5 years old and 6–10 years old, respectively) and ST ($p = 0.002$ and $p < 0.001$; 3–5 years old and 6–10 years old, respectively). However, overall, sleep duration (SD) increased ($F_{(1, 44)} = 4.63$, $p = 0.037$, $\eta2 = 0.10$). Notably, the prevalence rate increased in the younger age group ($p = 1$) and remained constant in the older age group ($p = 1$) (refer to Table 3).

According to maternal reports, the prevalence rates of children meeting all three movement guidelines declined during the coronavirus pandemic ($F_{(1, 44)} = 6.11$, $p = 0.017$, $\eta2 = 0.12$). The two age categories exhibited a decrease in meeting the guidelines (from 4.3% to 0% for the 3–5 age group and from 26.1% to 4.4% for the 6–10 age group). Additionally, the results revealed a significant main effect of age group ($F_{(1, 44)} = 6.82$, $p = 0.012$, $\eta2 = 0.13$), modulated by a significant interaction of the combined three 24-h movement guidelines X age group ($F_{(1, 44)} = 6.11$, $p = 0.017$, $\eta2 = 0.12$). The analysis suggests that the interaction arises from significant distinctions between the two age categories before the onset of the coronavirus pandemic (refer to Table 3).

Importantly, because there was a significant range in maternal education, and for results' robustness, we account for maternal education as a covariate. However, the analyses yielded the same pattern of results.

### 4. Discussion

The primary objective of this study was to examine the impact of the COVID-19 outbreak on health behaviors and adherence to the 24-h movement guidelines among Arab

Israeli children diagnosed with autism spectrum disorder (ASD). Additionally, the study aimed to explore potential variations between preschool and primary-school children. A secondary goal was to provide insights and contribute to the understanding of the consequences of the COVID-19 outbreak on children with neurodevelopmental disabilities.

### 4.1. Physical Activity

The results of this investigation align with previous research that has observed a reduction in physical activity participation among both typically developing children [3,7,28,30–34] and those with ASD [35–38]. Similar decreases have been noted in meeting physical activity recommendations for both healthy children [7,28,39,40] and those diagnosed with ASD [41–44]. The decline in physical activity was evident across various age groups, indicating a significant and concerning downward trend in typical physical rates among children with ASD throughout the coronavirus pandemic [20,44–47].

Social restrictions, including the shift to online education and treatment, along with home quarantine recommendations, have restricted the participation and engagement of ASD children in physical sessions, sporting, or physical activities related to the school environment. Consequently, the reduction in outdoor play may contribute to a decline in overall physical activity levels [22,48,49].

Prior to the pandemic outbreak, approximately 26% of children with ASD met the recommended daily physical activity guidelines, and this figure has declined to approximately 6.5%. In line with these results, a study by [7] reported that, before the pandemic, nearly 60% of typically developing Arab children met the recommended daily physical activity guidelines, and this has decreased to about 20%. The consistent inability of most children with ASD to meet the recommended physical activity guidelines aligns with findings from previous research [50–53]. For instance, supporting our results, Stanish et al. [54] observed that children with ASD spent significantly less time engaging in moderate–vigorous physical activity compared to their typically developing counterparts (29 min/day vs. 50 min/day, respectively), falling well below the recommended 60 min/day physical activity guidelines.

Our results reveal similar trends of significantly reduced physical activity (PA) for both age groups; however, older children experienced a more pronounced decline in PA during the COVID-19 outbreak (26.1% to 8.7% for the younger children and 26.1% to 4.4% for the older children). This finding contrasts with observations in typically developing children, where during COVID-19, older children (5–13 years old) were reported to more frequently meet the PA movement guidelines, and the decline in PA was more prominent in children aged 3–4 years old [7]. In our study, we observe a more severe decline in moderate–vigorous physical activity (MVPA) for children with ASD due to home confinement. A potential explanation for these detected MVPA differences might be related to the early atypical sensory sensitivity to stimuli observed in children with ASD. As they grow older, these children may encounter challenges in participating in activities with physical demands and social communication [54,55].

### 4.2. Screen Time

The findings also indicate an increase in screen time (ST) for both preschool children and school-age children during the COVID-19 pandemic. These results align with a substantial body of literature reporting on the effects of the COVID-19 pandemic and home confinement on both typically developing children [3,7,28,56–58] and children with ASD [41,45,59–61]. Yet, in contrast to the outcomes presented by [7], the surge in screen time was conspicuously more pronounced within the ASD sample.

The current study's findings support the "structured day hypothesis", a phenomenon suggesting that children and young people exhibit healthier habits, such as increased physical activity and decreased screen time, when following a structured timetable during the day [62]. This implies that the disruption of structured education days could be a crucial factor explaining the reduction in physical activity and increase in screen time. Given that individuals with ASD often thrive in an established routine and structure,

this disturbance to their daily lives may adversely impact their health behaviors [63,64]. A deeper exploration of how the "Structured Days Hypothesis" may elucidate the impact of the COVID-19 pandemic on the health behaviors of children and youth with ASD is warranted [20,62].

Moreover, the context of the COVID-19 pandemic and lockdown restrictions may provide additional instances supporting the "displacement theory" [28]. According to this theory, screen time and physical activity participation time could be interchangeable. By the ages of eight and ten, children typically become less active, leading to increased sedentary behaviors and greater weight gain [65]. Another potential explanation for the current findings could be the challenges mothers face in restricting sedentary behaviors, such as screen time, in their children [39]. Studies have shown a link between children's screen time and decreased mental well-being, with prolonged screen time being associated with psychological manifestations like decreased self-control, emotional instability, and signs of depression. [66,67]. These findings regarding physical activity and screen time suggest that both behaviors should be considered as priorities for health behavior interventions in children with ASD.

The results indicate that the preschool group tended to meet screen time guidelines less than schoolchildren, both before and during the outbreak of the pandemic. This trend may be attributed to the fact that, for young children with ASD, barriers to physical activity are more frequent and seem to be more complex than those encountered by typically developing (TD) children or older children with ASD. Particularly, the social and behavioral impairments knowledgeable by younger children with ASD seem to make involvement in structured and unstructured forms of physical activity more difficult [68]. Preschool children need greater play in outdoor spaces, and when they are less involved in play and physical activity, they have greater access to devices with screens, particularly during lockdowns, as they spend a significant amount of time at home. In contrast, schoolchildren may be more engaged in educational activities derived from online learning and instruction and have fewer barriers to being engaged in physical activity. Another contributing factor could be that educated mothers, who may have to work from home, allow their children to engage in screen-based games [69].

*4.3. Sleep Duration*

The current survey's outcomes reveal an increase in sleep duration and the proportion of children meeting sleep suggestions throughout the coronavirus outbreak in both age categories. These results are consistent with those of [7] and [28], who reported improvements in sleep duration and adherence to sleep guidelines among typical Arab Israeli, Spanish, and Brazilian children. Similarly, children in Canada were found to experience longer sleep periods during the COVID-19 pandemic [33].

It is worth noting that the small sample size of the current study may have limited the ability to detect differences in sleep patterns. While our study's results cannot pinpoint the exact reasons for the lack of change in sleep duration, the shift from in-person to online teaching/learning might have provided additional time for sleep. In contrast to our findings, Bruni et al. (2021) [60] reported no significant increase in sleep duration among children and adolescents with ASD (24.1% increase; 25% decrease) during the COVID-19 pandemic. Garcia et al. (2021) [46] also found no changes in sleep duration among adolescents with ASD during the COVID-19 outbreak compared to the period before. The discrepancies in results may be attributed to differences in sample size, other sleep difficulties experienced by ASD participants, and the consideration of additional aspects of sleep quality and sleep cycle changes in these studies.

While the current study indicates a positive shift in children's sleep patterns during the COVID-19 pandemic, future research should explore other aspects of sleep, such as sleep quality and the potential for changes in the sleep cycle. Determining whether the rise in sleep hours is linked to compromised sleep quality remains a complex task, a hypothesis that warrants investigation in subsequent studies. Notably, a higher percentage

of participants in the 6–10 age group adhered to the recommended amount of sleep. It is recommended that children aged 3 to 4 sleep 10-to-13 h a day, and children aged 6 to 10 sleep 8-to-11 h a day [29]. In both age groups, the percentage of children complying with the sleep guidelines increased accordingly.

The results of the current study indicate that the percentage of children meeting the combined 24-h movement guidelines decreased from 15.2% before the COVID-19 pandemic to 2.2% during the pandemic. In contrast, a study by [7] reported a decline from 12% before the pandemic to 3.1% during the COVID-19 outbreak, indicating a more significant decrease in the ASD sample.

These findings underscore the impact of the COVID-19 pandemic on the overall movement behaviors of children with ASD. The substantial reduction in the percentage of children meeting the combined guidelines suggests a significant disruption in their PA, ST, and SD during the COVID-19 outbreak. Further research and interventions may be necessary to address these challenges and promote healthy behaviors among children with ASD, especially during periods of extended disruptions, like the coronavirus outbreak.

The findings of the current survey emphasize the importance of promoting healthy daily routines and behaviors for children and youth during and after pandemic restrictions, particularly for those with neurodevelopmental disabilities. As a result, it is recommended to boost physical activities and reduce screen time during periods of "homestay" to improve overall health outcomes in children and youth.

Parents, educators, health councils, and authorities should be attentive to this situation and make concerted efforts to develop practical strategies and interventions aimed at increasing physical activity levels and preventing negative health-related behaviors to the greatest extent possible. Implementing these measures can contribute to the well-being and overall health of children and youth, especially during challenging circumstances, such as the COVID-19 pandemic.

### 4.4. Limitations and Implications for Future Studies

The present research is subject to certain limitations that warrant consideration. Firstly, due to health risks, face-to-face interactions were avoided, and reliance on virtual self-reporting introduces potential weaknesses compared to in-person assessments.

Secondly, mothers' assessments of their children's functioning might be influenced by the mothers' own well-being; however, the study focused exclusively on child behavior and did not collect data regarding the mothers' own experiences with COVID-19. Future research endeavors should focus on the children and specifically examine their performance throughout the COVID-19 pandemic.

Thirdly, generalizing the sample to other populations may be challenging, as it was drawn exclusively from the Arab Israeli population. To investigate potential cultural influences, future studies could compare the effects of COVID-19 on Arab and Jewish children in Israel.

While a larger sample size would enhance the study's effectiveness, logistical constraints in the current setup posed challenges in collecting greater samples on a broader scale.

In future experiments, it is advisable to explore potential mediating factors related to parents' environments and socioeconomic status, which may affect parental attention, availability, and the ability to address their children's needs and challenges.

The study does not contain information about the IQ and autistic-symptom factors that can affect the physical activity and screen time of the children. Furthermore, we acknowledge the possibility that children with mild ASD may have had different experiences with the COVID-19 restrictions compared to those with more severe symptoms. Therefore, future studies should address this issue by exploring potential variations in experiences based on the severity of ASD symptoms and other relevant factors and their effects on physical activity and screen time with a larger sample size.

Therefore, the effect of these factors on the physical activity and screen time of autistic children should be considered in future studies.

Lastly, parents provided information about their children's behavior both before and during the pandemic. However, the passage of time might have introduced bias into their reports. Specifically, there could be bias in reporting children's behavior prior to the pandemic.

## 5. Conclusions

In conclusion, this study's findings contribute to the growing body of knowledge about the adverse effects of the COVID-19 pandemic on the health-related behaviors and movement patterns of children with ASD. Children and their families encounter numerous challenges due to COVID-19 and the associated physical isolation restrictions, and these challenges may be more pronounced for children with neurodevelopmental disorders like ASD. Consequently, it becomes crucial to emphasize health-promoting behaviors, interventions, and strategies tailored to children's needs.

Understanding the impact of COVID-19 on health behaviors in children and young adults with ASD is vital for developing effective strategies to mitigate negative effects and guard against potential long-term health risks from the current pandemic and similar events in the future, such as wars. Authorities, scientific communities, regional healthcare organizations, and legislators hold primary responsibility for devising and implementing targeted interventions to protect children and enhance their well-being and mental health. Collaboration among educators, social policy experts, administrators, and parents is essential to reduce the detrimental effects of social distancing and school closures, thereby fostering children's development, health, and overall well-being.

**Funding:** This research received no external funding.

**Institutional Review Board Statement:** The study was conducted in accordance with the Declaration of Helsinki, and approved by the Ethic Committee of Oranim College of Education (protocol code: 103, December 2020).

**Informed Consent Statement:** Informed consent was obtained from all participants involved in the study.

**Data Availability Statement:** The data presented in this study are available from the corresponding author upon request.

**Conflicts of Interest:** The author declares no conflict of interest.

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
