# Peer review of "Changes in Healthy Behaviors among Arab Israeli Children Diagnosed with ASD amid the Coronavirus Outbreak: Mothers’ Perceptions"

_education, doi:10.3390/educsci14030253_

Round 1
Reviewer 1 Report
Comments and Suggestions for Authors
The paper satisfies all the criteria of a scientific paper. The literature review is pertinent and gives the theoretical background of the study. The method is satisfactory although the sample is small. The research tool is adequate. The analysis is going step by step leading to several conclusions. There are also limitations and implications for future studies. The study contributes to the growing body of knowledge about the adverse effects of the COVID-19 pandemic on the health-related behaviors of ASD children, a special population which needs special care. The researchers seem to be experienced both in researching and writing. No other suggestions apart from line 208 where the title of the sub-chapter should go to the next line. Congratulations for the excellent work.
Reviewer 2 Report
Comments and Suggestions for Authors

Round 2
Reviewer 2 Report
Comments and Suggestions for Authors
Lines 44-62: It is crucial for the authors to focus their attention on elucidating the characteristics of the Arab population, as highlighted in the subsequent bolded lines, instead of delving into sociological issues pertaining to discrimination. This is particularly relevant considering the researchers' acknowledgment that only a portion of the Arab population perceives discrimination. The inclusion of such references is confusing and detracts from the article's central focus, warranting their omission. Furthermore, it doesn't answer the question of whether the entire Arab population shares the same characteristics.
"Approximately 85% of the Arab population in Israel identifies as Muslim Arabs, while 7.4% are Druze, and 7.2% are Christian Arabs, according to the Central Bureau of Statistics [CBS] 2019. Among the Muslim Arab population, 16% are Bedouins, comprising about 3.5% of the total Israeli population. Furthermore, nearly 90% of the Arab population resides in predominantly Arab towns, primarily located in northern Israel. The Negev region in the south of Israel is home to two-thirds of the Bedouin population, with some residing in unrecognized settlements lacking basic infrastructure such as electricity, sewage systems, and telephone lines (as reported by CBS 2019).
Lines 198-202: The authors should state the geographical region, northern Israel, or the Negev region. According to the introduction section, disparities exist between the Arab population residing in the country's northern region and those in the Negev region.
